# Yield Stress and Reversible Strain in Titanium Nickelide Alloys after Warm Abc Pressing

**DOI:** 10.3390/ma12193258

**Published:** 2019-10-06

**Authors:** Aleksander Lotkov, Victor Grishkov, Anatoly Baturin, Victor Timkin, Dorzhima Zhapova

**Affiliations:** Institute of Strength Physics and Materials Science of the Siberian Branch of the Russian Academy of Sciences, Tomsk, 634055, Russia; grish@ispms.tsc.ru (V.G.); abat@ispms.tsc.ru (A.B.); timk@ispms.tsc.ru (V.T.); dorzh@ispms.tsc.ru (D.Z.)

**Keywords:** martensitic transformations, superelasticity, shape memory, yield stress, plastic strain, abc pressing, torsion deformation

## Abstract

The results of the position analysis of the yield stress τ_0.3_ on the "stress–strain" (τ–γ) dependences, received at the torsion of specimens of Ti_49.8_Ni_50.2_ (at%) alloy are presented. The critical stress τ_0.3_ (IV), corresponding to the end of linear stage III and the beginning of the intensive development of plastic strain at stage IV, preceding the fracture of the specimens, were obtained as well. The structure of the specimens was transformed from coarse-grained to microcrystalline as a result of warm (723 K) abc pressing with a true deformation *e* of 8.4. The regularities of the development of reversible inelastic strain (superelasticity, SE, and shape memory effect, SME) and plastic strain γ_pl_ after isothermal (295 K) loading of specimens up to τ ≤ τ_0.3_(IV), unloading, and their subsequent heating up to 500 K are studied. From the joint analysis of the “τ–γ” dependences obtained at 295 K and "plastic strain–total strain" dependences the yield stress τ_0.3_ corresponding to the development of 0.3% of the plastic strain under loading of the specimens was determined. Critical stress τ_0.3_(IV) was determined as equal to the stress corresponding to a deviation of 0.3% from the linear “τ–γ”dependence at stage III. It is shown that the yield stress τ_0.3_ for all specimens is localized at the beginning of stage III for all specimens. The ratio τ_0.3_(IV)/τ_0.3_ is from 2.3 to 3.8. The accumulation of plastic strain at stage III (after loading with τ from τ_0.3_ to τ_0.3_(IV)) is from 2.4% to 4.7% (depending on the true deformation of the specimens during warm abc pressing). Thus, stage III is the stage of deformation hardening of specimens under torsion. On the basis of the results of this and previous works it is shown that, in alloys with thermoelastic martensitic transformations and with thermomechanical memory, the ratio τ_0.3_(IV)/τ_0.3_ can vary in a wide range: in reinforced specimens τ_0.3_ can be close to τ_0.3_(IV), and in more ductile specimens τ_0.3_ can be significantly less than τ_0.3_(IV). However, in order to correctly determine the yield stress of τ_0.3_ and the corresponding strain γ_t_(0.3), it is necessary to carry out a joint analysis of “τ–γ” and "plastic strain–total strain" dependencies.

## 1. Introduction

TiNi-based materials, being bright representatives of smart materials, are widely used in engineering and medicine due to their high level of superelasticity (SE) and shape memory effect (SME), strength and plasticity, thermomechanical and thermal fatigue life, and corrosion resistance and biocompatibility. The excellent thermomechanical properties of TiNi are provided by thermoelastic martensite transformations from a cubic B2 phase to a rhombohedral R or a monoclinic B19′ phase. The shape of specimens with low internal stress remains unchanged under cooling and heating over the temperature ranges of martensitic transformations because of the formation of polivariant systems of self-accommodated martensite domains. SE and SME are conditioned by the deformation influence. Shape memory is the ability for accumulation of reversible inelastic strain in isothermal loading–unloading cycles at *T*_d_ < *M*_f_ (where *T*_d_ is the deformation temperature and *M*_f_ is the martensite finish temperature) and for its recovery in unloaded states of specimens at *T* > *A*_f_ (where *A*_f_ is the austenite finish temperature). Superelasticity is the ability for accumulation of inelastic strain during the formation of stress-induced martensite (SIM) under isothermal loading at stresses higher than the martensite shear stress σ_M_ (or τ_M_ in torsion) and for its complete recovery under further unloading due to reverse SIM→B2 transition. The deformation temperature *T*_d_ for SE realization lies within the so-called superelastic window: Δ*T* = *T*_2_ – *T*_1_ with *M*_s_ < *T*_1_ < *T*_d_ < *T*_2_ < *M*_d_, where *M*_s_ is the martensite start temperature on cooling and *M*_d_ is the maximum SIM formation temperature under isothermal loading. Another important property of this type of alloy is the ability to produce high reactive stresses on the heating of specimens preliminary deformed in martensite (*T*_d_ < *M*_s_) or SIM (*T*_d_ > *M*_s_) states and constrained to prevent shape recovery.

In general, the degree of shape recovery η, being the ratio of reversible to total strains, and the reactive stress depend on development of plastic strain at the stages of preliminary loading and unloading or heating after predeformation of alloy specimens with thermoelastic martensitic transformations. Conventionally, the onset of plastic deformation in a material is judged from its macroscopic yield stress, i.e., from the stress σ_0.2_ (in compression and tension) or τ_0.3_ (in torsion) at which the accumulated plastic strain reaches 0.2% and 0.3%, respectively. The higher the stress σ_0.2_ compared to σ_M_, the higher the SME, the SE, and the reactive stresses. The parameter Δσ = σ_0.2_ – σ_M_ is analyzed in many studies [1,2,3,4,5], showing its significance on the way to better performance and wider applicability of TiNi alloys. The yield stress of materials can be increased by different methods: variation in their elemental composition (e.g., alloying), thermomechanical treatment [5,6,7], and severe plastic deformation with hardening via grain structure refinement to fine- and ultrafine-grained states [8,9,10,11]. However, correctly determining the yield stress of alloys like TiNi is problematic. The stress–strain diagram of materials with no thermoelastic martensite transformations includes a linear elastic stage followed by a plastic stage, and their yield stress criterion is the stress σ_0.2_ or τ_0.3_ at which the dependence deviates from linearity by 0.2% (tension, compression) or by 0.3% (torsion) in the transition region between the elastic and plastic stages. For alloys with thermoelastic martensite transformations, such as TiNi, the stress–strain diagram is more complicated (Figure 1).

The diagram begins with short elastic stage I which passes to stage II at stresses σ > σ_M_ (τ > τ_M_). Stage II represents martensite detwinning and reorientation (*T*_d_ < *M*_f_) or SIM generation (*T*_d_ > *A*_f_) responsible for inelastic strain accumulation at comparatively small stresses. Then comes stage III in which the response to increasing stresses is almost linear and the strain hardening coefficient θ = dσ/dε (dτ/dγ) is higher than that in stage II. As σ > σ_0.2_(IV) is reached, stage IV begins showing intense plastic flow with further fracture of specimens. In general, the total strain ε_t_ at any of stages II–IV comprises elastic martensite strains ε_el_, reversible inelastic strains ε_SE_ and ε_SME_, and plastic strains ε_pl_: ε_t_ = ε_el_ + ε_SE_ + ε_SME_ + ε_pl_. Thus, the dependences σ–ε and τ–γ allow us to correctly determine σ_M_ and τ_M_, but determining σ_0.2_ and τ_0.3_ from these dependences is impossible without analysis of the reversible strain ε_el_ + ε_SE_ + ε_SME_ and plastic strain ε_pl_ with increasing the total strain ε_t_.

Another concept is also available to judge the position of σ_0.2_ on stress–strain diagrams [1,3,4,12,13,14,15,16,17,18]. The concept suggests that stage III corresponds to elastic martensite strains, and the yield stress to σ_0.2_(IV) and τ_0.3_ (IV) (Figure 1). The idea of such an interpretation has apparently come from a study of Cu–Zn–Si alloys [19] in which almost complete strain recovery was found under unloading at total strains ε_t_ close to the end of stage III, suggesting that stage III is elastic for the martensite formed at stage II (*T*_d_ > *M*_s_). The idea was then extended to Cu-, Ag-, and Au-based alloys with thermoelastic martensite transformations [20,21]. Similar judgments were made for TiNi-based alloys [1] from an analysis of their reversible inelastic and plastic behavior [22]: in loading–unloading cycles with heating to *T* > *A*_f_ in unloaded states, TiNi alloys rich in C and Si completely recovered their shape at ε_t_ ≤ 8% with maximum reactive stresses at ε_t_ ≈ 8% late in stage III (note that ε_L_ was also introduced as a critical strain above which plastic deformation begins in a material [1]). In further studies [14,15,16,17,18], the stress σ_0.2_ in TiNi-based alloys was assessed from the criterion of 0.2% deviation from linearity at stage III.

However, even early studies reported that after tension to ε_t_ at the end of stage II, unloading, and heating in unloaded states, the plastic strain was ~1% [23] in binary alloys with a Ni content of 50.1 at% (initial B19′ martensite). It was noted [24] that in stage III, the slopes (dσ/dε or dτ/dγ) of “stress–strain” dependences under loading and unloading is essentially different for specimens of TiNi-based alloys. This contradicts the supposition that only elastic deformation of reoriented martensite occurs at stage III. Additionally, a high dislocation density was found by transmission electron microscopy in TiNi deformed at stage III [24]. These results were confirmed by further studies [25,26,27,28,29,30,31,32,33,34]. The finish of reorientation and detwinning of martensite domains (in specimens with initial martensite structure) and formation of SIM (in specimens with initial B2 structure), the appearance of high dislocation density and formation of compound twins, such as {20?1}_B19__′_, {110}_B19__′_, and {1?13}_B19__′_, were observed after deformation at stage III of different TiNi-based alloys with polycrystalline [25,26,27,28,29,30,31] and monocrystal [32,33,34] structures. Consequently, the development of both inelastic strains and plastic strain may develop at stage III.

In several studies [25,26,29,30,32,33,34], the ε_t_ dependences of ε_el_, ε_SE_, ε_SME_, and ε_pl_ were analyzed in isothermal cycles of loading and unloading with further heating to complete shape recovery via reverse martensite transformation to B2. The plastic strain at ε_t_ close to the end of stage III was shown to measure 1–5%, but the yield stress σ_0.2_ from the σ–ε and ε_pl_–ε_t_ dependences was not determined. One of the recent studies [35] provides data on the γ_t_ dependences of γ_SE_, γ_SME_, and γ_pl_ in coarse-grained and microcrystalline Ti_49.2_Ni_50.8_ (at%) under torsion in isothermal τ–γ cycles (295 K) with heating in unloaded states. The τ–γ and γ_pl_–γ_t_ dependences of Ti_49.2_Ni_50.8_ (with an average grain size of 43 µm to 1.5 µm after warm caliber rolling at 723 K) show that τ_0.3_ lies near the beginning of stage III and that after deformation with γ_t_ corresponding to τ_0.3_(IV), the plastic strain is 1% irrespective of the average grain size in the material. The difference Δτ = τ_0.3_(IV) – τ_0.3_ is considerable and increases from 160 MPa in the coarse-grained material to 460 MPa in the microcrystalline one.

Here we analyze the yield stress τ_0.3_, plastic strain, and reversible inelastic strain under torsion in Ti_49.8_Ni_50.2_ (at%) with an average grain size of 40 µm to ~1.5 µm after warm abc pressing at 723 K.

## 2. Materials and Methods

The almost equiatomic alloy Ti_49.8_Ni_50.2_ at%, containing Ti_4_Ni_2_ (N,O) in an amount of ~6 vol%, was supplied as hot-swaged bars of diameter 20 mm (MATEK-SMA, Moscow, Russia). After high-temperature abc forming at 1023 K, its specimens were shaped as cubes with an edge of ~20 mm. The specimens were exposed to warm multicycle abc pressing in a die at 723 K with upsetting in three orthogonal directions during each cycle. The strain rate was $\dot{\varepsilon }$ = 0.16–0.18 s^–1^, and the total true plastic strain per cycle was *e* = 0.2–0.3, being the sum of natural logarithm of ratios of specimen heights before and after upsetting. Before each upsetting event, the die with a specimen was heated in a furnace at 723 K. The specimen temperature during abc pressing varied by no more than 10 deg. In the initial state after hot forming at 1023 K, the specimens had a coarse-grained structure with an average grain size <d> = 40 µm. As the strain *e* was increased, the grain size <d> decreased, measuring 1.5 µm after warm abc pressing with *e* = 8.4 at 723 K. A detailed discussion of the effect of pressing at 723 K on the grain-subgrain structure of the alloy can be found elsewhere [36]. After warm abc pressing at 723 K, the true strain in the specimens was 0.3, 0.6, 1.8, 4.2, 6.4, and 8.4.

The inelastic and plastic strains in the material under torsion was studied on an inverted torsion pendulum with an operating temperature of 573–120 K. The test specimens were bars of diameter ~1 mm with a gage length of ~10 mm. The experimental method used to determine the reversible inelastic strains γ_SE_, γ_SME_ and plastic strain γ_pl_ is shown in Figure 2. The total strain is γ_t_ = γ_SID_ + γ_pl_ = γ_el_ + γ_SE_ + γ_SME_ + γ_pl_. The “τ–γ” dependencies for each strain γ_t_ were obtained in isothermal (295 K) “loading-unloading” cycles. The value of γ_SE_ (the superelasticity effect) was assumed to be equal to the recovery of the strain under isothermal unloading (including a small Hook strain of ~1.5% [37]): γ_SE_ = γ_t_ – γ_r_. After unloading, the martensite phase was partially present providing the presence of residual strain γ_r_ in the unloaded specimens at 295 K. The martensitic phase transforms into the B2 phase with the subsequent heating of the unloaded specimens. This martensite transformation provides the recovery of inelastic strain during heating: γ_SME_ = γ_r_ – γ_pl_. The plastic strain γ_pl_ is equal to the residual strain after completion of the shape recovery. The value of γ_pl_ for all specimens was determined at 500K (~150 degrees above A_f_ in the initial specimens). The strains γ_t_, γ_r_, and γ_pl_ strains are equal to *arctg*S_t_, *arctg*S_r_, and *arctg*S_pl_, where $S_{t}=\left(d\phi_{t}\right)/2l$, $S_{r}=\left(d\phi_{r}\right)/2l$, $S_{pl}=\left(d\phi_{pl}\right)/2l$; ϕ_t_, ϕ_r_, and ϕ_pl._ are the torsional angles of the specimens in radians under isothermal loading with γ_t_, after complete unloading and at 500 K, respectively; d and l are the dimensions of the cross-section and the length of the specimen gage section. In each next cycle γ_t_ was increased. The final value of γ_t_ is equal to γ_t_(IV), corresponding to τ_0.3_(IV) of each specimen with the true strain *e* after warm abc pressing (Figure 1). The degree of the shape recovery of the specimens was determined to be η = γ_SID_ / γ_t_.

The sequence and temperatures of martensite transformations in the alloy after warm abc pressing at 723 K were considered previously [36]. Of importance here is the following aspect: In Ti_49.8_Ni_50.2_ (at%) preliminarily quenched from 1073 K, only B2↔B19′ transitions occur on cooling and heating. The martensite start and finish temperatures in the quenched alloy are *M*_s_ = 317 K and *M*_f_ = 219 K (B2→B19′ transition), and the austenite start and finish temperatures are *A*_s_ = 343 K and *A*_f_ = 354 K (reverse B19′→B2 transition). The initial alloy preformed at high temperature undergoes B2→R→B19′ transitions on cooling. The rhombohedral R phase is formed at *T*_R_ = 327 K, and the temperatures of R→B19′ and reverse B19′→B2 transitions lie in the range from *M*_s_ = 316 K to *M*_f_ = 281 K and from *A*_s_ = 338 K to *A*_f_ = 355 K, respectively. Thus, the temperatures of B2→B19′, R→B19′, and B19′→B2 in the alloy after quenching and after high-temperature forming are almost the same.

After warm abc pressing with *e* = 0.3–8.4 at 723 K, the sequence of martensite transformation in the alloy on cooling and heating remains unchanged: B2→R→B19′ and B19′→B2, respectively. The temperatures *T*_R_, *A*_s,,_ and *A*_f_ remain constant, and *M*_s_ and *M*_f_ decrease linearly to 311 K and 276 K (by ~5 and ~10 deg, respectively) with an increase in *e* to 8.4. Thus, both the initial and the abc-pressed specimens have a two-phase R + B19′ structure at 295 K.

## 3. Experimental Results

Figure 3 shows the engineering stress–strain diagrams of the initial and abc-pressed TiNi specimens under isothermal loading at 295 K. It is shown that the dependences comprise stages I–IV typical for alloys with thermoelastic martensite transformations (Figure 1).

Since both the initial and abc-pressed specimens at 295 K have a two-phase R + B19′ structure (*T*_d_ > *M*_f_), the deformation stage II in both involves the formation of SIM due to their R→B19′ transition under loading and the reorientation and detwinning of the B19′ martensite present before loading. The martensite shear stress τ_M_ as a function of the initial true strain *e* is shown in Figure 4.

Figure 4 shows that the martensite shear stress τ_M_, or the pseudo-yield stress, decreases noticeably after pressing with *e* ≤ 0.6 and remains constant with further increasing *e* to 8.4. As can be seen from Figure 3, the pseudo-yield stage gradually passes to stage III at which the strain increases linearly with the applied stress. At this stage, the strain hardening coefficient θ = dτ/dγ increases with *e* specified in warm abc pressing (Figure 3). The stress τ_0.3_(IV), corresponding to a deviation of 0.3% from the previous linear stage III, causes the material to enter stage IV, which culminates in its fracture. In Figure 3, the stress τ_0.3_(IV) in the initial specimen and specimens pressed with *e* = 1.8 and *e* = 8.4 is 710, 870, and 915 MPa, respectively.

To correctly determine the yield stress τ_0.3_, which corresponds to an accumulated plastic strain of 0.3%, we should analyze the dependence of the plastic strain γ_pl_ on the total strain γ_t_ in torsion. All components of reversible inelastic strains γ_SID_ = γ_SE_ + γ_SME_ and γ_pl_ as a function of γ_t_ obtained in the study are for isothermal loading–unloading cycles at 295 K with heating in unloaded states to 500 K (as described in the Materials and Methods). As can be seen from Figure 5, the γ_t_ dependences of γ_pl_ for all specimens are qualitatively similar. Increasing the total strain γ_t_ to ~16% slightly increases the plastic strain γ_pl_, and as γ_t_ goes above 16%, γ_pl_ begins to grow. The data in Figure 3 show that γ_t_ = 16% is close to γ_t_(IV), corresponding to τ_0.3_(IV). Thus, the critical stress τ_0.3_(IV) in the initial and abc-pressed specimens characterizes the onset of intense plastic flow at stage IV rather than the onset of yielding, i.e., the yield stress τ_0.3_.

The γ_t_ dependences of γ_pl_ allow us to rather easily determine the yield stress τ_0.3_ on τ–γ diagrams like those in Figure 3. From the γ_t_ dependences of γ_pl_ we can estimate the total strain γ_t_(0.3) at which γ_pl_ = 0.3%, and the values of γ_t_(0.3) will give the position of the yield stress τ_0.3_ on the τ–γ diagrams. Figure 6 and Figure 7 show the yield stress τ_0.3_ and strain γ_t_(0.3), and the stress τ_0.3_(IV) and strain γ_t_(IV) in the initial and abc-pressed specimens. 

From Figure 6 it is seen that τ_0.3_(IV) remains almost unchanged after abc pressing with *e* ≤ 0.6 and increases as *e* is increased to 1.8. In the specimens pressed with *e* = 1.8–8.4, the increment in τ_0.3_(IV) is ~200 MPa. The diagrams in Figure 7 show that the strain γ_t_(IV) corresponding to the end of stage III and the transition to stage IV decreases by 0.4% after abc pressing with *e* = 4.2 compared to γ_t_(IV) in the initial specimens. After pressing with *e* = 6.4 and *e* = 8.4, the decrease in γ_t_(IV) is ~1%. By and large, all τ_0.3_(IV) correspond to the range of γ_t_(IV) from 17% to 18.5%. The yield stress τ_0.3_ in the specimens pressed with *e* = ≤ 0.6 at 723 K decreases to 220 MPa against its value τ_0.3_ = 300 MPa in the initial specimens. Consequently, the work softening is observed in the specimens after warm abc pressing with *e* ≤ 0.6. After abc pressing with *e* = 0.6–8.4, the yield stress τ_0.3_ increases linearly almost to its initial value. The secondary hardening of specimens is the result of warm pressing with *e* > 0.6. Thus, the yield stress τ_0.3_ in the initial (coarse-grained) and abc-pressed (microcrystalline structure) specimens differs little, measuring 300 and 320 MPa, respectively. As can be seen from Figure 7, the *e* dependence of γ_t_(0.3), corresponding to τ_0.3_, differs from that of γ_t_(IV). 

The initial alloy reaches τ_0.3_ at γ_t_(0.3) = 7%. After abc pressing with *e* = 0.6, the yield stress τ_0.3_ shows its minimum (Figure 6) at γ_t_(0.3) = 4.5% (Figure 7), and as *e* is increased from 0.6 to 4.2, the value of γ_t_(0.3) grows to that of the initial alloy. After abc pressing with *e* = 4.2–8.4, the increment in γ_t_(0.3) slows down to ~0.5%. As a whole, the variation in γ_t_(0.3) is ~3%, which is twice as large as the variation in γ_t_(IV) after pressing with *e* up to 8.4. The extent of stage III in the initial alloy is 11.5%, reaches its maximum of ~14% after pressing with *e* = 0.6, and decreases to 9.5% after pressing with *e* = 8.4. 

The plastic strain γ_pl_(IV) after loading to τ_0.3_(IV) can be estimated from a joint analysis of the τ–γ and γ_pl_–γ_t_ dependences in Figure 3 and Figure 5. Figure 8 shows the plastic strain γ_pl_(IV) as a function of *e* specified in warm abc pressing.

Figure 8 shows that, against the decrease in τ_0.3_ and almost invariant τ_0.3_(IV) in the specimens pressed with *e* ≤ 0.6, their plastic strain γ_pl_(IV) increases from 3.6% to its maximum 4.7% at *e* = 0.6. The specimens pressed with *e* > 0.6, in which the yield stress τ_0.3_ grows, reveal a monotonic decrease in γ_pl_(IV). However, even in the specimens with *e* = 6.4 and 8.4, the plastic strain accumulated in torsion at γ_t_ = γ_t_(IV) corresponding to τ_0.3_(IV) remains high (~2.4%). Thus, the actual yield stress in Ti_49.8_Ni_50.2_ is close to the onset of stage III on its τ–γ dependences (Figure 3). Stage III is a strain hardening stage at which γ_pl_ increases from 0.3% to 2.4–4.7%. 

Figure 9 and Figure 10 show the total reversible inelastic strains γ_SID_(0.3) and γ_SID_(IV) recovered after isothermal loading to γ_t_(0.3) and γ_t_(IV), which correspond to τ_0.3_ and τ_0.3_(IV), unloading, and heating in unloaded states as a function of *e* in abc pressing (γ_SID_ = γ_SE_ + γ_SME_ = γ_t_ –γ_pl_). As can be seen, the dependences are qualitatively similar. In the specimens pressed with *e* ≤ 0.6, the strains γ_SID_(0.3) and γ_SID_(IV) are smaller than those in the initial specimens. The strains γ_SID_(0.3) and γ_SID_(IV) increase after abc pressing with *e* from 0.6 to 4.2. After pressing with *e* from 4.2 to 8.4, the strain γ_SID_(0.3) increases somewhat while γ_SID_(IV) decreases somewhat. The dependence of γ_SID_(0.3) is close to that of γ_t_(0.3) because the accumulated plastic strain is low (0.3%) and is the same for all specimens. The behavior of γ_SID_(IV) is governed by variations of both γ_t_(IV) (Figure 7) and γ_pl_(IV) accumulated under loading to τ_0.3_(IV) (Figure 8).

The minimum value of γ_SID_(IV) after abc pressing with *e* = 0.6 is due to the maximum value of γ_pl_(IV) and to the constancy of γ_t_(IV). After pressing with 0.6 < *e* < 4.2, the plastic strain γ_pl_(IV) decreases more greatly than γ_t_(IV), and this allows γ_SID_(IV) to reach its maximum almost equal to γ_SID_(IV) in the initial specimens. When γ_pl_(IV) and γ_t_(IV) are decreased simultaneously, the specimens pressed with *e* = 6.4 and 8.4 show a slight decrease in γ_SID_(IV). 

An important characteristic of shape memory alloys is the degree of shape recovery η = (γ_t_ – γ_pl_)/γ_t_ = γ_SID_/γ_t_, being the ratio between reversible inelastic strains produced under loading to γ_t_ and recovered under unloading (SE) and further heating (SME). Figure 11 shows the degree of shape recovery η_0.3_ and η_0.3_(IV) after loading to τ_0.3_ and τ_0.3_(IV), respectively, as a function of *e* in warm abc pressing. After abc pressing with *e* ≤ 0.6, η_0.3_ decreases from 96% to 93%, and after abc pressing with *e* from 1.8 to 8.4, its value is η_0.3_ = 96% as in the initial specimens.

The *e* dependences of η_0.3_ and η_0.3_(IV) are qualitatively similar. However, η_0.3_(IV) is 10–20% lower than η_0.3_ both in the initial and in the abc-pressed specimens. This is because plastic strain develops under loading at stage III (Figure 8). In the initial specimens, η_0.3_(IV) = 80%. In the specimens pressed with *e* ≤ 0.6, it decreases to 74% and, in those pressed with *e* from 1.8 to 8.4, it increases to 85%. Thus, in terms of complete shape recovery, the range of total strains γ_t_ ≤ γ_t_(0.3) at applied stresses τ < τ_0.3_ is more preferable.

Another important result concerns the total strain γ_SID_ = γ_SE_ + γ_SME_. From comparison of Figure 9 and Figure 10 it follows that γ_SID_(IV) is two times higher than γ_SID_(0.3). The inelastic strains γ_SE_(0.3), γ_SME_(0.3) and γ_SE_(IV), γ_SME_ (IV) for loading to τ_0.3_ and τ_0.3_(IV) are plotted in Figure 12. 

As can be seen from Figure 12, γ_SME_ > γ_SE_ in all specimens irrespective of their grain structure and total strains γ_t_(0.3), γ_t_(IV). The strain γ_SME_(0.3) is close to γ_SID_(0.3) because γ_SE_(0.3) is small and is almost equal to the recovery of elastic martensite strain under unloading: 1–1.5% [37]. Increasing the total strain under loading to τ_0.3_(IV) increases γ_SE_(IV) to 3–4% in the initial specimens and specimens after the pressing. The strain γ_SME_(IV) in these specimens weakly depends on *e*, and its value γ_SME_(IV) = 10–11% is 2–3 times higher than γ_SME_(0.3). Thus, coarse-grained and microcrystalline Ti_49.8_Ni_50.2_ structures plastically deformed by torsion are promising SME materials capable of providing high reversible inelastic strains after loading with τ_0.3_(IV).

## 4. Discussion

Data on yield stress in alloys with martensitic transformations, including TiNi, are necessary for assessing their operability and reliability as structures with SE- and SME-based functional properties. In our analysis, we follow two approaches to yield stress estimations in near-equiatomic Ti_49.8_Ni_50.2_ (at%) transformed from coarse-grained to microcrystalline states by warm abc pressing at 723 K. The first approach implies an analysis of the τ–γ dependence (Figure 1) with linear stage III taken as elastic martensite deformation and yield stress as stresses at which stage III ends and passes to stage IV. The second approach implies a joint analysis of the τ–γ dependence and γ_pl_–γ_t_ dependence in loading–unloading cycles at increasing γ_t_ with the yield stress taken as stress τ_0.3_, at which the accumulated plastic strain reaches 0.3%. 

Our study shows that in Ti_49.8_Ni_50.2_ alloy, whether coarse-grained or microcrystalline after warm pressing, the accumulated plastic strain at stage III under torsion with τ = τ_0.3_(IV) ranges from 2.4% to 4.7%, i.e., it is much higher than 0.3%. Assuming that τ_0.3_(IV) is the yield stress, the degree of shape recovery after loading to τ_0.3_(IV), unloading, and further heating, would measure η_0.3_(IV) = (97.3 ± 0.1)% in all specimens irrespective of their structure. However, its actual value after loading to τ = τ_0.3_(IV) varies from 74% to 85% depending on *e*. The stress τ_0.3_(IV) is 700–920 MPa, and the yield stress τ_0.3_ compared to τ_0.3_(IV) is 2–3 times smaller, measuring 230–320 MPa and approximating the onset of stage III. After such cycles with loading to τ = τ_0.3_, the degree of shape recovery remains high: 93–96%. 

These results are qualitatively similar to the results of our previous study [35] in which we analyzed the yield stress τ_0.3_ and critical stress τ_0.3_(IV) under torsion in Ti_49.2_Ni_50.8_ (at%) alloy transformed from coarse- to fine-grained states by warm caliber rolling at 723 K. For the alloy, which had a two-phase B2+R structure before torsion at *T*_d_ = 295 K, the ratio τ_0.3_(IV)/τ_0.3_ was 1.45 ± 0.20 depending on the true strain in rolling. The stress τ_0.3_ was also close to the onset of stage III, but the degree of shape recovery after loading, unloading, and heating to 500 K in Ti_49.2_Ni_50.8_ alloy was higher than its value in Ti_49.8_Ni_50.2_: 97% and 93% after loading to τ_0.3_ and τ_0.3_(IV), respectively. The plastic strain accumulated at stage III with τ = τ_0.3_(IV) was 1%. 

Thus, our research in abc-pressed Ti_49.8_Ni_50.2_ (at%) and caliber-rolled Ti_49.2_Ni_50.8_ (at%) [35] demonstrates that jointly analyzing the dependences τ–γ and γ_pl_–γ_t_ in loading, unloading, and heating cycles allows one to estimate the yield stress τ_0.3_, critical stress τ_0.3_(IV), plastic strain accumulated at stage III, and total inelastic strain γ_SID_ and its components γ_SE_ and γ_SME_ depending on the total torsional strain γ_t_.

It is seen from Figure 10 and Figure 12 that, in Ti_49.8_Ni_50.2_ alloy after warm abc pressing, the inelastic strains γ_SID_, γ_SE_, and γ_SME_ increase 2–2.5 times after loading with total strains from γ_t_(0.3) to γ_t_(IV) at stage III. The dependences of γ_SID_, γ_SE_, and γ_SME_ are similar to those obtained for the same material in torsion with γ_t_ > γ_t_(IV) at *T*_d_ = 295 K [38]: after deformation with γ_t_ > γ_t_(IV), the strains γ_SID_, γ_SE_, and γ_SME_ continue to grow, reach their maximum values, and then decrease. However, the increments in γ_SID_, γ_SE_, and γ_SME_ are much lower than that after loading with γ_t_ from γ_t_(0.3) to γ_t_(IV). The maximum value of γ_SID_ equal to 16–18% is reached in the range γ_t_ = 27–48% with the plastic strain measuring γ_pl_ = 10–30%. The increment in γ_SME_ is no greater than 1–2%: γ_SME_ increases to 11–13%. The increase in γ_SID_ to its maximum is governed mostly by γ_SE_, which increases from 4% after deformation at γ_t_ = γ_t_(IV) to 6–7% at γ_t_ = 27–48%. Thus, ~85% of the reversible inelastic strain in Ti_49.8_Ni_50.2_ alloy is attained after loading with γ_t_ corresponding to stage III on the τ–γ dependence. 

In summary, it should be noted that, in Ti_49.8_Ni_50.2_ alloy after warm abc pressing, the ratio τ_0.3_(IV)/τ_0.3_ is 2.4–3.8 and, in Ti_49.2_Ni_50.8_ alloy after warm rolling, it is 1.45 ± 0.2 [35]. At the same time, research data are available on two TiNi-based alloys in which the yield stress and the critical stress at the end of stage III are very close [14,16]. After compression at 293 K and unloading, complete shape recovery was observed in single-crystal Ti_48.5_Ni_51.5_ (at%) alloy aged at 823 K for 1.5 h [14]. Polycrystalline Ti_50.2_Ni_49.8_ (at%) alloy hardened by a single pass of equal channel angular pressing (channel angle 90°) at room temperature revealed its complete shape recovery after isothermal (293 K) cycles of compression with strains close to the end of stage III and unloading with further heating [16].

Thus, the ratio τ_0.3_(IV)/τ_0.3_ or σ_0.2_(IV)/σ_0.2_ can vary over a wide range depending on the composition of an alloy and its preliminary thermomechanical treatment. However, the yield stress in alloys with martensitic transformations, including TiNi, can be correctly determined only from a joint analysis of stress–strain dependences (τ–γ or σ–ε) and dependences of accumulated plastic strains on total strains (τ_pl_–γ_t_ or σ_pl_–σ_t_). The similarity of the methods for determining the actual yield stress under torsion, compression, and tension of TiNi-based alloy specimens is provided by the following results. The deformation behavior of these alloys is qualitatively similar for these deformation modes. The same four deformation stages (I–IV) are observed on the “σ–ε” dependences obtained under tension and compression of TiNi-based alloy specimens, as well as on their “τ–γ” dependences obtained under torsion. The development of plastic ε_pl_ strain and reversible inelastic strains (SE, SME, total inelastic strain) were studied and depend on the total ε_t_ tension strain in isothermal "loading–unloading" cycles with subsequent heating of the unloaded specimens of binary alloys with 50.5 at% Ni (polycrystalline structure [29]) and 50.6 at% Ni (single-crystal with [100] orientation [31]). In [29,31] it was noted that the onset of ε_pl_ development corresponded to the transition from stage II (pseudo-yield "plateau") to stage III, but σ_0.2_ was not determined in these studies. By the end of stage III ε_pl_ reached 2.5% in polycrystalline specimens [29] and ~5% in single-crystal specimens [31]. At the same time, the maximum total reversible inelastic strain was attained after loading at stresses corresponding to the end of stage III. Similar studies of the plastic strain ε_pl_ and reversible inelastic strains were carried out in [32] under compression of single-crystal (orientation [001]) specimens of the TiNi(Mo,Fe) alloy with the initial B2 structure. The results [32] showed that 0.2% of the plastic strain appears after loading with σ_0.2_ = 580 MPa (the beginning of stage III on the “σ–ε” dependence). After compression with σ ≈ 900 MPa (the end of stage III and the transition to stage IV) the plastic strain increased up to 2%. Consequently, σ_0.2_(IV) / σ_0.2_ ≥ 1.6 in these specimens. In general, the results [29,31,32] for TiNi-based alloy specimens under compression and tension are qualitatively similar to those obtained for Ti_49.8_Ni_50.2_ (at%) alloy specimens after warm abc pressing in our study and for Ti_49.2_Ni_50.8_ (at%) alloy specimens after warm rolling under torsion [35].

In conclusion, the following should be noted: Non-monotonous changes of yield stress τ_0.3_, intensity of plastic strain accumulation, and corresponding changes of reversible inelastic strains under torsion are determined by the work softening and further strengthening of specimens during warm abc pressing. The physical causes for these processes are not discussed in our paper because no experimental data are presently available for such a discussion. At present, research is conducted on the fine crystalline and dislocation structures in the material after warm abc pressing and after torsion. The results of this research will be presented in our future publications.

## 5. Conclusions

Our study shows that in Ti_49.8_Ni_50.2_ (at%) transformed from coarse-grained to microcrystalline states by warm abc pressing at 723 K the yield stress τ_0.3_ on its τ–γ diagram occurs early in stage III featuring a linear increase in torsional strains under applied stress. The end of stage III and the onset of stage IV, with its intense plastic flow and eventual fracture, fall on the critical stress τ_0.3_(IV) corresponding to a deviation of 0.3% from the previous linear τ–γ stage. The ratio τ_0.3_(IV)/τ_0.3_ in the alloy is 2.3–3.8.

At total strains in the range from τ_0.3_ to τ_0.3_(IV), the accumulated plastic strain in the abc pressed alloy is 2.4–4.7%. Hence, stage III represents strain hardening.

After isothermal torsional loading to τ_0.3_ and unloading at 295 K with further heating to 500 K, the degree of shape recovery in the TiNi alloy is 93–96% both in its initial state and in its states after abc pressing with different true strains *e*. The total inelastic strain γ_SID_ in such TiNi specimens is no greater than ~7% and is mostly due to the component γ_SME_ while the component γ_SE_ approximates the elastic strain and measures ~1.5%. As the applied stress is increased from τ_0.3_ to τ_0.3_(IV), the degree of shape recovery in the TiNi specimens after loading, unloading, and heating decreases to 74–85% while the total reversible inelastic strain grows to 14–15% of which 10–11% is recovered via heating (shape memory effect) and ~4% via unloading (superelasticity). Thus, at τ_0.3_(IV), both components γ_SME_ and γ_SE_ increase, but γ_SE_ remains much lower than γ_SME_. Such plastically deformed specimens with high inelastic strain recovery may be useful for manufacture of functional elements with shape memory.

Reasoning from this study and studies reported elsewhere [14,16,35], we can conclude that the ratio τ_0.3_(IV)/τ_0.3_ in alloys with martensite transformations is widely variable: in hardened specimens, the yield stress τ_0.3_ can approximate τ_0.3_(IV), and its value in more plastic specimens can be much lower than τ_0.3_(IV). By and large, the yield stress τ_0.3_ and the total strain γ_t_(0.3) for its reach should be assessed by jointly analyzing of “stress–strain” and “plastic strain–total strain” dependences.

## Figures and Tables

**Figure 1 materials-12-03258-f001:**
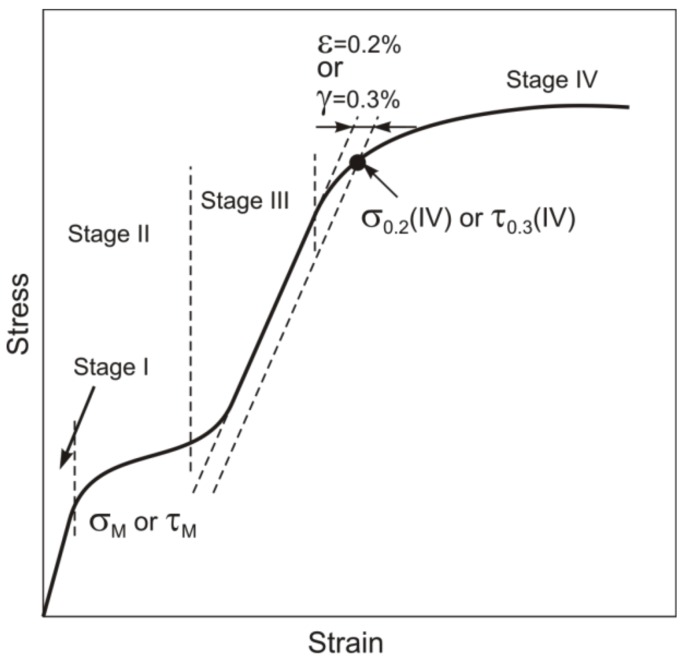
Schematic stress–strain diagram with stages I–IV at temperatures *T*_d_ < *M*_d_ for alloys with martensitic transformations: σ_M_, τ_M_: martensitic shear stresses; σ_0.2_(IV), τ_0.3_(IV): critical stresses are equal to yield stress σ_0.2_, τ_0.3_ (according to [1,3,4,12,13,14,15,16,17,18]).

**Figure 2 materials-12-03258-f002:**
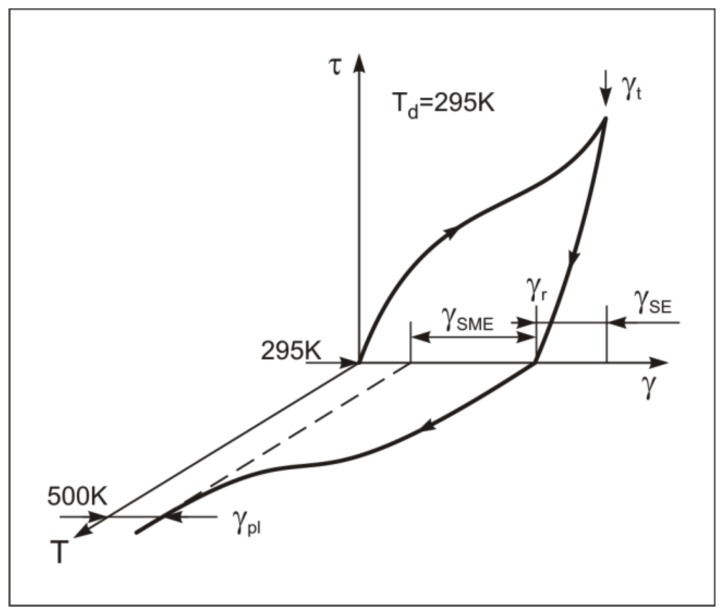
The accumulation and recovery of strain in isothermal (T_d_ = 295 K) cycle “τ-γ” and during the subsequent heating of unloaded specimens (scheme).

**Figure 3 materials-12-03258-f003:**
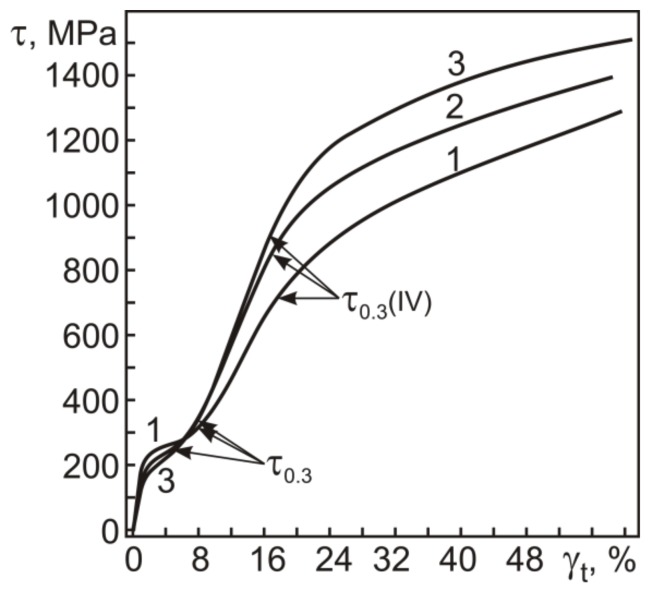
Engineering stress–strain diagrams in torsion at 295 K for Ti_49.8_Ni_50.2_ before (1) and after warm abc pressing at 723 K with true strains *e* = 1.8 (2) and *e* = 8.4 (3).

**Figure 4 materials-12-03258-f004:**
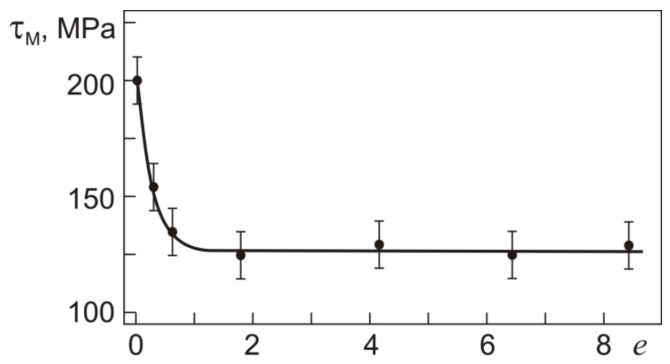
Martensite shear stress τ_M_ vs. true strain *e* in T_49.8_Ni_50.2_ in torsion at 295 K.

**Figure 5 materials-12-03258-f005:**
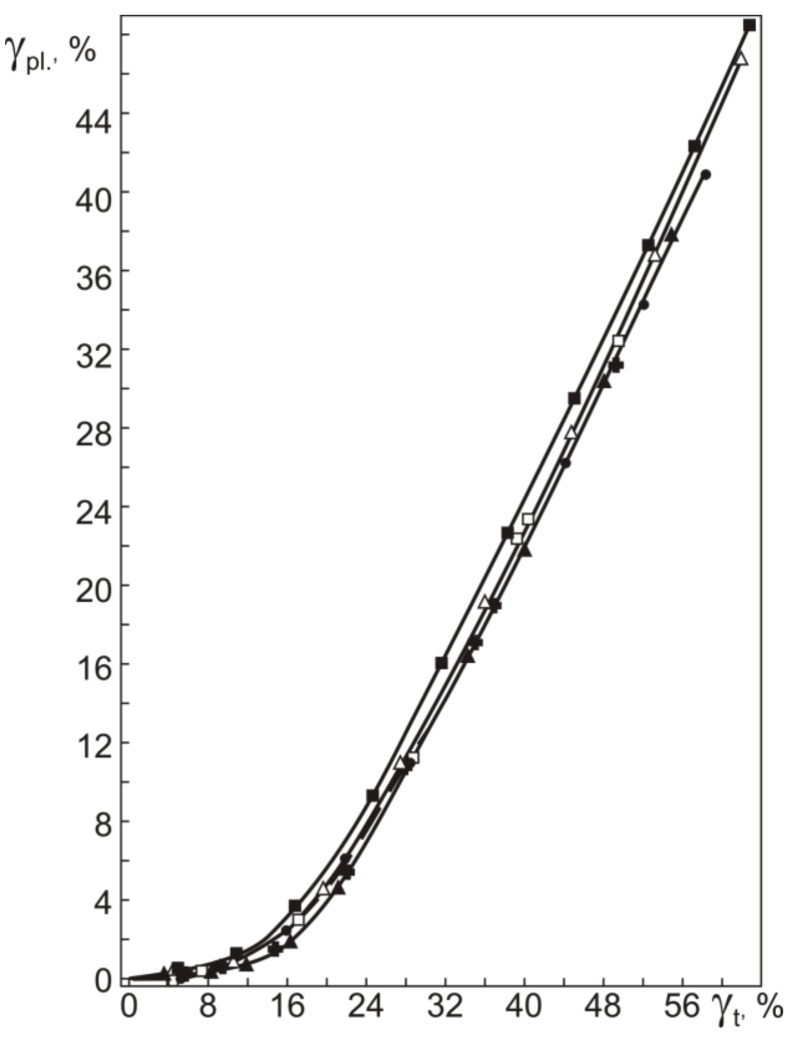
Plastic strain γ_pl_ vs. total strain γ_t_ in torsion at 295 K for Ti_49.8_Ni_50.2_ before (●) and after abc pressing with *e* equal to 0.6 (■), 1.8 (△), 4.2 (☐), 6.4 (✚), and 8.4 (▲).

**Figure 6 materials-12-03258-f006:**
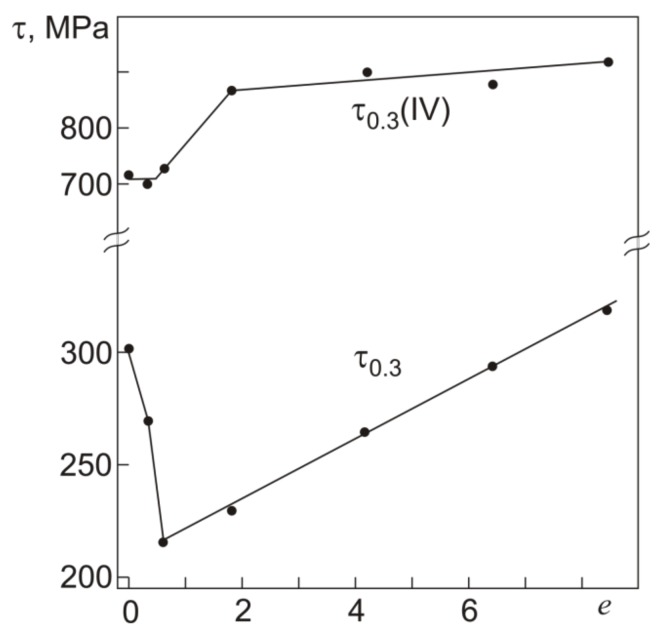
Yield stress τ_0.3_ and critical stress τ_0.3_(IV) in torsion at 295 K for Ti_49.8_Ni_50.2_ after abc pressing with different true strains *e*.

**Figure 7 materials-12-03258-f007:**
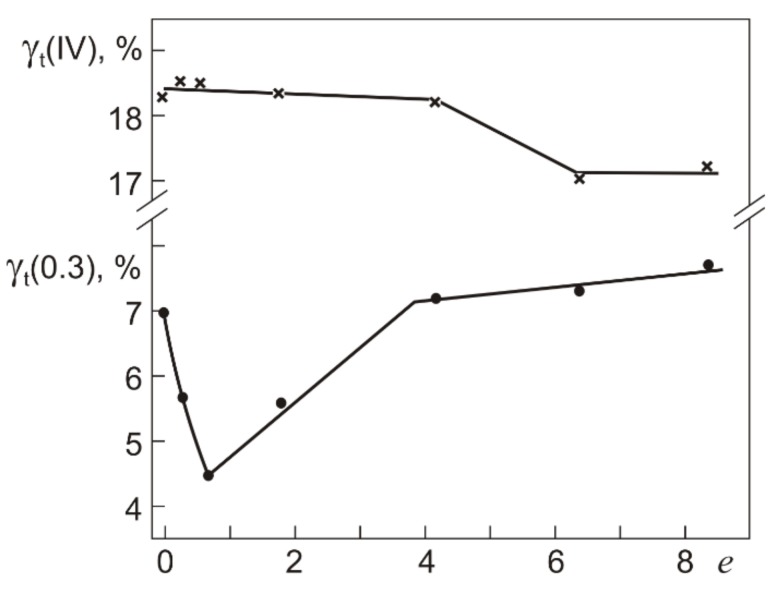
Total strains γ_t_(0.3) and γ_t_(IV) at which τ_0.3_ and τ_0.3_(IV) are reached in Ti_49.8_Ni_50.2_ after warm abc pressing with different true strains *e*.

**Figure 8 materials-12-03258-f008:**
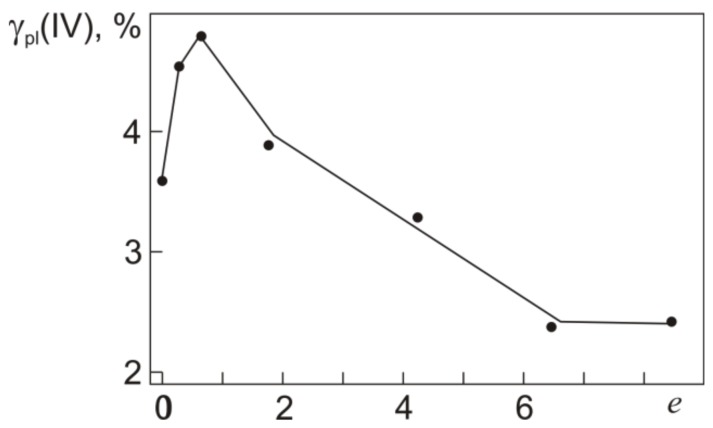
Plastic strain γ_pl_(IV) corresponding to τ_0.3_(IV) early in stage IV in torsion at 295 K for Ti_49.8_Ni_50.2_ after abc pressing with different true strains *e*.

**Figure 9 materials-12-03258-f009:**
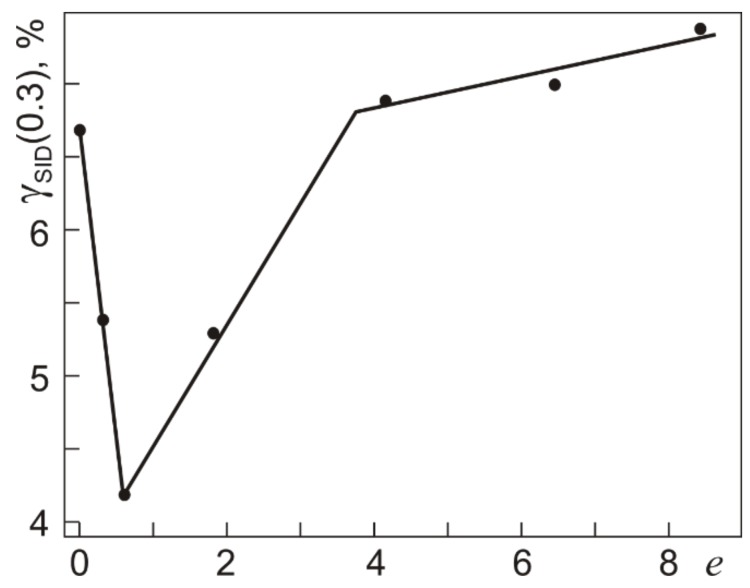
Total reversible inelastic strain γ_SID_(τ_0.3_) recovered by Ti_49.8_Ni_50.2_ after torsion with τ = τ_0.3_ (295 K) and heating (500 K) at different true strains *e* specified in warm abc pressing. Residual plastic strains γ_pl_ = 0.3%.

**Figure 10 materials-12-03258-f010:**
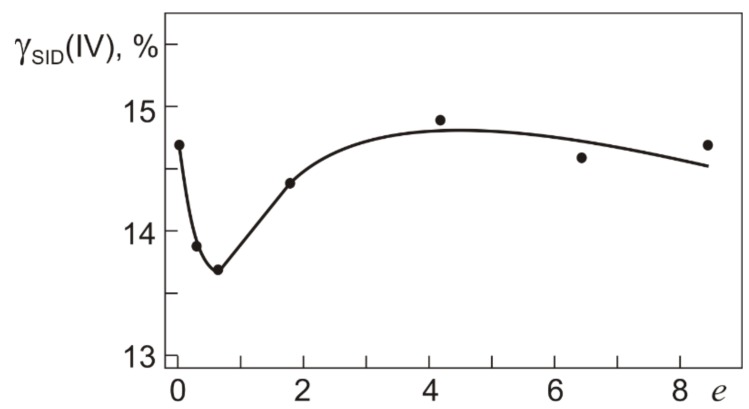
Total reversible inelastic strain γ_SID_(IV) recovered by Ti_49.8_Ni_50.2_ after torsion with τ = τ_0.3_(IV) (295 K) and heating (500 K) at different true strains *e* specified in warm abc pressing.

**Figure 11 materials-12-03258-f011:**
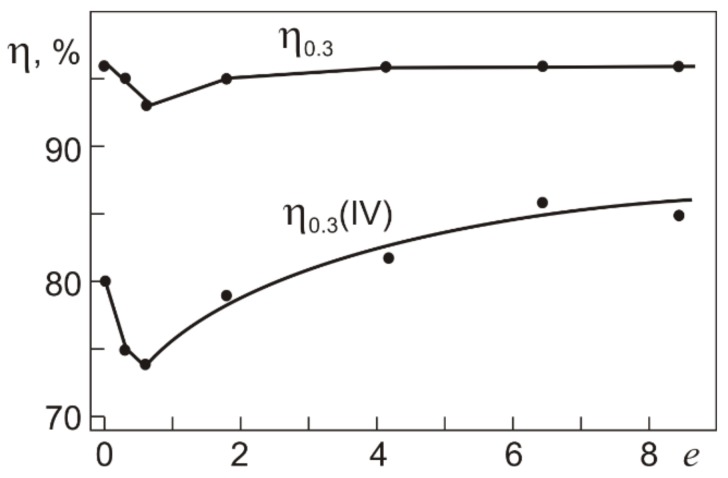
The degree of shape recovery η_0.3_ at τ_0.3_ and η_0.3_(IV) at τ_0.3_(IV) in Ti_49.8_Ni_50.2_ in isothermal loading–unloading cycles (295 K) with heating in unloaded states (500 K) as a function of the true strain *e* in warm abc pressing.

**Figure 12 materials-12-03258-f012:**
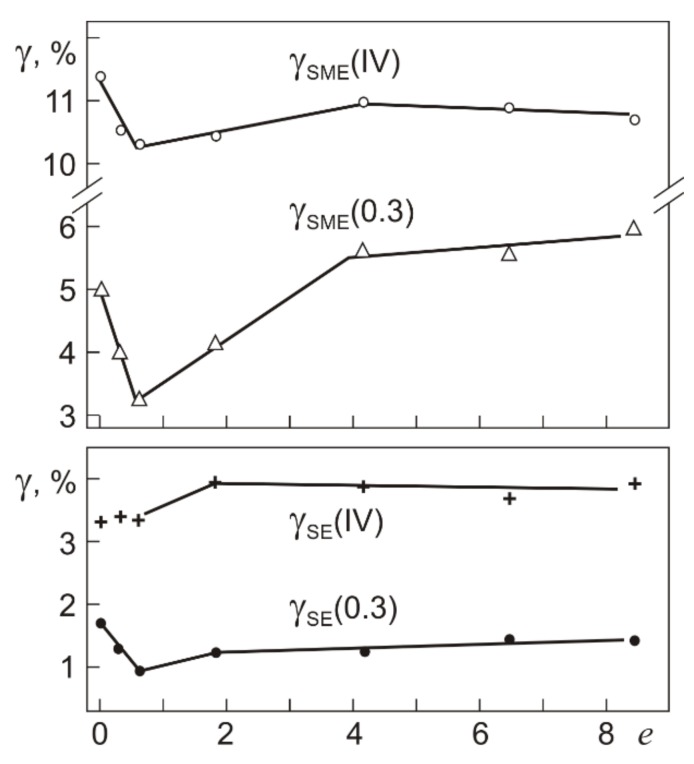
Reversible inelastic strains γ_SE_(0.3) and γ_SME_(0.3) at τ_0.3_, and γ_SE_(IV) and γ_SME_(IV) at τ_0.3_(IV) in Ti_49.8_Ni_50.2_ in isothermal loading–loading cycles (295 K) with heating in unloaded states (500 K) as a function of the true strain *e* in warm abc pressing.

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
