# Peer review of "Yield Stress and Reversible Strain in Titanium Nickelide Alloys after Warm Abc Pressing"

_materials, 2019, doi:10.3390/ma12193258_

Round 1

Reviewer 1 Report

In this paper, the authors essentially extrapolate inelastic strains and critical values for their evolution from stress–strain diagrams obtained from torsion of NiTi bars under isothermal loading. Subsequent heating of the unloaded samples is needed for distinguishing between the inelastic deformations that take place during torsion. The obtained results are compared for different NiTi microstructures of the same composition.

The results are sound and sufficiently described, and of importance for calibration and validation of models. Regarding the technological importance of the paper, it is limited to structural applications of NiTi that may experience high loads that are rather limited. Regarding the insight offered to the microstructure-property relation, I believe that further discussions that go beyond a mere description of the results are in order. For example, should one expect the yield stress to increase when the grain size decreases or not and why? Why the recoverable strain magnitude increases in one vs the other case. Is it because of the yield stress alone or the value of the maximum transformation strain as well?

Author Response

Point 1: Regarding the insight offered to the microstructure-property relation, I believe that further discussions that go beyond a mere description of the results are in order. For example, should one expect the yield stress to increase when the grain size decreases or not and why? Why the recoverable strain magnitude increases in one vs the other case. Is it because of the yield stress alone or the value of the maximum transformation strain as well? 

Response 1: The authors agree that the questions proposed by the reviewer for discussion are interesting and important. At the same time, it should be noted that in order to discuss them correctly, experimental results are needed that are not available at present. In particular, it is known that the most effective increase in yield stress is observed in titanium nickel-based alloys with submicrocrystalline and nanocrystalline structures. In our samples of Ti49.8Ni50.2 (at%) alloy even after pressing at 723K with e from 6.4 to 8.4 a microcrystalline structure with an average grain size of 1.5 microns is formed. The portion of grains/subgrains of submicrocrystalline sizes is small. In addition, as a result of warm pressing, a non-monotonous change in the deformation behavior of the alloy is observed. After pressing with e ≤ 0.6, when the specimen structure is coarse-grained, the alloy becomes softer (τ0.3 decreases). The secondary hardening of the alloy after pressing with e > 0.6 correlates with an effective transition from coarse-grained to microcrystalline structure. However, the influence of grain size changes under pressing and the influence of other physical factors (change of dislocation structure during pressing, formation of new twins during torsion, etc.) on yield stress and critical stress τ0.3(IV) has not been studied at present and is not the purpose of this paper.

About the factors with influence on the value of reversible inelastic strain. Correlation of the value of reversible inelastic deformation with the processes of strengthening and work softening, which determine the development of plastic strain, is shown in the paper. At the same time, we do not discuss the physical factors determining the increase of reversible inelastic strain at the increase of the total strain at the stage III under torsion. In [29, 31] it was suggested that the increase in reversible inelastic strain at stage III may be due to the completion of reorientation and detwining of martensite B19′ domains or the completion of the formation of stress-induced martensite (depending on the deformation temperature). It is possible that in the samples of Ti49.8Ni50.2 (at%) alloy studied in our work, the increase in reversible inelastic strain at stage III may be due to these causes. However, at present, discussion of the influence of all these factors on the development of plastic strain and reversible inelastic strain is impossible without experimental data on the fine crystal and dislocation structures of the samples both after warm abc pressing and under torsion. We noted this in the corresponding section of the text (Line 387-394 of revision manuscript).

Reviewer 2 Report

This article presents the experimental results of the critical stresses at two defined stages of deformation of NiTi alloy under torsion and corresponding shear strain values after different levels of warm ABC pressing. 

There are plenty of typing errors within the text. 

Line 17: ... loading of samples up to τ ... (typing error).

Line 19: check this line for correctness.  

Line 30: What does it mean by "the true yield stress of 0.3"?

Line 31: check this line for the typing errors.

Line 36: "TiNi based, being ..." to be replaced by "TiNi based materials, being ...".

Line 50: Td is called strain temperature, however, it was defined as the deformation temperature in Line 45.

Line 56-58: check it correctness.

Line 84: In the equation for ε_t, the first term should be ε_el.

Line 106: ... TiNi-based poly-crystals [25–31] and single-crystals [32–34]) ...: "(" is missing.  

In "Materials and Methods": the experimental setup especially how to measure the recoverable strain need to be explained.

Line 143-150: needs to be rewritten to be clearer for the reader.

Line 167: "Experimental and results" to be replaced by "Experimental results"

Line 187: According to Figure 2, the stress τ0.3(IV) is above 700 MPa for the three cases, thus 300, 230, and 320 MPa are not right.

For describing Figures 7, 8 and 9: in addition to what is observed from each figure, the scientific reason for this variation in material behaviour must be presented.

In "Discussion" or other sections: clarify if understanding the behaviour of NiTi under torsion (or pure shear stress) can be used for understanding the behaviour of this alloy under other loading conditions. 

Line 373: "... in smart alloys ..." to be replaced by "... in shape memory alloys ...".     

Author Response

Point 1: There are plenty of typing errors within the text.

Response 1: Many typing errors appear due to the change of font from “Times New Roman” to the font used by editorial office “Palatino Linotype”. The typing errors are corrected. The corrections are marked by green color.

Point 2: Line 17: ... loading of samples up to τ ... (typing error).

Response 2: Lines 16-17: ... γpl after isothermal (295K) loading of samples up to τ≤τ0.3(IV)...

Point 3: Line 19: check this line for correctness.

Response 3: Lines 18-19: …of the “τ-γ”dependences obtained at 295K and "plastic strain – total strain" dependences

Point 4: Line 30 and Line 31: What does it mean by "the true yield stress of 0.3"? 

Response 4: Line 30: ... the yield stress of τ0.3 and the corresponding strain γt(0.3), it...

Point 5: Line 36: "TiNi based, being ..." to be replaced by "TiNi based materials, being ...".

Response 5: Line 37: TiNi-based materials, being …

Point 6: Line 50: Td is called strain temperature, however, it was defined as the deformation temperature in Line 45.

Response 6: Line 51: The deformation temperature Td for SE realization…

Point 7: Line 56-58: check it correctness.

Response 7: Lines 55-60: …(Td < Ms) or SIM (Td > Ms) states and constrained to prevent shape recovery.

In general, the degree of shape recovery η, being the ratio of reversible to total strains, and the reactive stress depend on development of plastic strain at the stages of preliminary loading and unloading or heating after predeformation of alloy specimens with thermoelastic martensitic transformations.

Point 8: Line 84: In the equation for ε_t, the first term should be ε_el.

Response 8: Line 86: …εt = εel + εSE + εSME + εpl

Point 9: Line 106: ... TiNi-based poly-crystals [25–31] and single-crystals [32–34]) ...: "(" is missing.

Response 9: Lines 108-112: The finish of reorientation and detwinning of martensite domains (in specimens with initial martensite structure) and formation of SIM (in specimens with initial B2 structure), the appearance of high dislocation density and formation of compound twins such as {20‾1}B19′, {110}B19′, {1‾13}B19′ were observed after deformation at stage III of different TiNi-based alloys with polycrystalline [25–31] and monocrystal [32–34] structures.

Point 10: In "Materials and Methods": the experimental setup especially how to measure the recoverable strain need to be explained. Line 143-150: needs to be rewritten to be clearer for the reader.

Response 10: Line 143-161 and Figure 2: This paragraph is rewritten. We include the Figure 2, which illustrates the determination of inelastic recoverable strain and plastic strain.

Lines 143-161: The experimental method for determining of reversible inelastic strains γSE, γSME and plastic strain γpl is shown in Figure 2. Total strain γtSIDplelSESMEpl. The “τ-γ” dependencies for each strain γt were obtained in isothermal (295K) “loading-unloading” cycles. The value of γSE (the superelasticity effect) was assumed to be equal to the recovery of strain at isothermal unloading (including a small Hook strain of ~1.5% [37]): γSE = γt – γr. After unloading, the martensite phase remains partially, which causes the presence of residual strain γr in the unloaded samples at 295K. The martensitic phase transforms into B2 phase at the subsequent heating of the unloaded samples. This MT causes the recovery of inelastic strain during heating: γSMEr–γpl. The plastic strain γpl is equal to the residual strain after the completion of the shape recovery. The value of γpl for all samples was determined at 500K (~150 degrees above Af in the initial samples). The γt, γr and γpl strains are equal to arctgSt, arctgSr and arctgSpl, where St=(dφt)/2l , Sr=(dφr)/2l , Spl=(dφpl)/2l ; φt, φr and φpl. - torsional angles of the samples in radians at isothermal loading with γt, after full unloading and at 500K, respectively; d and l - dimensions of cross-section and length of working part of samples. In each subsequent cycle γt was increased. The final value of γt is equal to γt(IV) corresponding to τ0.3(IV) of each specimen with true deformation e after warm abc pressing (Figure 1). The degree of shape recovery of the samples was determined as η=γSIDt.

Figure 2. (see in text of PDF file)

Figure 2. The accumulation and recovery of strain in isothermal (Td=295K) cycle “τ-γ” and during the subsequent heating of unloaded specimens (scheme)

Point 11: Line 167: "Experimental and results" to be replaced by "Experimental results"

Response 11: Line 177: “Experimental results”

Point 12: Line 187: According to Figure 2, the stress τ0.3(IV) is above 700 MPa for the three cases, thus 300, 230, and 320 MPa are not right.

Response 12: The error was removed. Lines 197-197: …In Figure 3, the stress τ0.3(IV) in the initial specimen and specimens pressed with е = 1.8 and e = 8.4 is 710, 870, and 915 MPa, respectively.

Point 13: For describing Figures 7, 8 and 9: in addition to what is observed from each figure, the scientific reason for this variation in material behaviour must be presented.

Response 13: We agree with reviewer’s remark that the determination of the physical factors leading to such deformation behavior of Ti49.8Ni50.2 (at.%) alloy is an important task. Figure 5 shows that the alloy specimens become softened after pressing with e≤0.6 (the increase of τ0.3). After pressing with e>0.6, the secondary hardening of specimens are observed (τ0.3 increases). This determines the changes in the value of plastic strain accumulation in the torsional process at 295K, resulting in corresponding changes in reversible inelastic strains and the degree of shape recovery, as shown in Figures 6-12. However, physical factors causing the work softening of specimens after pressing with low deformations e and their subsequent hardening after pressing with e>0.6 have not been identified at present. For this purpose it is necessary to carry out special studies of fine crystal and dislocations structure of samples both after abc pressing and its evolution under torsion. These studies are long and are carried out at present. The results of these studies can only be presented in future publications. For these reasons, the additional paragraph was introduced in text (Line 387-394).

Lines 387-394: In conclusion, the following should be noted. Non-monotonous changes of yield stress τ0.3, intensity of plastic strain accumulation and corresponding changes of reversible inelastic strains during torsion are determined by the work softening and subsequent strengthening of specimens during warm abc pressing. In this paper we have not discussed the physical reasons for these processes. The experimental data required for this are not available at present. A study of the fine crystalline and dislocation structures of samples both after warm abc pressing and samples deformed by torsion are carried out at present. The results of these studies will be presented in subsequent publications.

Point 14: In "Discussion" or other sections: clarify if understanding the behaviour of NiTi under torsion (or pure shear stress) can be used for understanding the behaviour of this alloy under other loading conditions.

Response 14: The additional paragraph was introduced in text to take into account this reviewer’s remark (Lines 364-386).

Lines 364-386: The similarity of the methods for determining of the actual yield stress under torsion, compression and tensile of TiNi-based alloy samples is provided by, the following results. The deformation behavior of these alloys is qualitatively similar for these deformation modes. The same 1-IV deformation stages are observed on “σ-ε” dependences obtained under tension and compression of TiNi-based alloy specimens as well as on “τ-γ” dependences obtained under torsion of similar specimens. The development of plastic εpl strain and reversible inelastic strains (SE, SME, total inelastic strain) were studied in depend on the total εt tension strain in isothermal "loading-unloading" cycles with subsequent heating of the unloaded specimens of binary alloys with 50.5 at.% Ni (polycrystalline structure [29]) and 50.6 at.% Ni (single-crystal with [100] orientation [31]). In [29, 31] it was noted that the beginning of εpl development corresponds to the transition from stage II (pseudo-yield "plateau") to stage III, but σ0.2 was not determined in these works. By the end of stage III εpl reached 2.5% in polycrystalline samples [29] and ~5% in single-crystal samples [31]. At the same time, the maximum total reversible inelastic strain was achieved after loading with a stress corresponding to the end of stage III. Similar studies of the development of εpl and reversible inelastic strains were carried out in [32] during compression of single-crystal (orientation [001]) samples of the TiNi(Mo,Fe) alloy with the initial B2 structure. The results [32] showed that 0.2% of plastic strain appears after loading with σ0.2=580 MPa (the beginning of stage III on the “σ-ε” dependence). After compression with σ≈900 MPa (finish of stage III and transition to stage IV) the plastic strain increased up to 2%. Consequently, σ0.2(IV)/σ0.2≥1.6 in these specimens. In general, the results [29, 31, 32] obtained by compression and tension of TiNi-based alloy specimens are qualitatively similar to those obtained by torsion of Ti49.8Ni50.2 (at.%) alloy samples after warm abc pressing in this work and Ti49.2Ni50.8 (at.%) alloy samples after warm rolling in [35].

Point 15: Line 373: "... in smart alloys ..." to be replaced by "... in shape memory alloys ...".

Response 15: Lines 412-414: Such plastically deformed specimens with high inelastic strain recovery may be useful for manufacture of functional elements with shape memory.

Responses to the reviewer's comments are repeated in the PDF file. Please see the attachment

Round 2

Reviewer 2 Report

The manuscript has been improved, however the following statement seems to be incorrect (Line 104-106): 

"It was noted that not only elastic martensite deformation occurs at stage III because most of the TiNi-based alloys after loading at stage III and unloading
feature hysteresis on their stress–strain curves [24]." 

Also, please check the rest of paragraph.  

Author Response

Point 1: The manuscript has been improved, however the following statement seems to be incorrect (Line 104-106):

"It was noted that not only elastic martensite deformation occurs at stage III because most of the TiNi-based alloys after loading at stage III and unloading feature hysteresis on their stress–strain curves [24]."

Also, please check the rest of paragraph.

Response 1: We take into account the Reviewer’s remark. The text of this paragraph was checked. The revision text is next.

Lines 104-107: ...It was noted [24] that in stage III, the slopes (ds/de or dt/dg) of stress–strain dependences under loading and unloading are essentially different for TiNi-based alloys. This contradicts the supposition that only elastic deformation of reoriented martensite occurs at stage III…

and

Lines 113-114: …Consequently, the development of both inelastic strains and plastic strain may develop at stage III…
